# RespiCoV: Simultaneous identification of Severe Acute Respiratory Syndrome Coronavirus 2 (SARS-CoV-2) and 46 respiratory tract viruses and bacteria by amplicon-based Oxford-Nanopore MinION sequencing

Annika Brinkmann[1]*, Steven Uddin[1], Sophie-Luisa Ulm[1], Katharina Pape[1], Sophie Förster[1], Khalid Enan[2], Jalal Nourlil[3], Eva Krause[1], Lars Schaade[1], Janine Michel[1], Andreas Nitsche[1]

1 Highly Pathogenic Viruses, Centre for Biological Threats and Special Pathogens, WHO Reference Laboratory for SARS-CoV-2 and WHO Collaborating Centre for Emerging Infections and Biological Threats, Robert Koch Institute, Berlin, Germany, 2 Ministry of Higher Education and Scientific Research, Khartoum, Sudan, 3 Pasteur Institute of Morocco, Casablanca, Morocco

* brinkmanna@rki.de

## Abstract

Since December 2019 the world has been facing the outbreak of the **Severe Acute Respiratory Syndrome Coronavirus 2** (SARS-CoV-2). Identification of infected patients and discrimination from other respiratory infections have so far been accomplished by using highly specific real-time PCRs. Here we present a rapid multiplex approach (RespiCoV), combining highly multiplexed PCRs and MinION sequencing suitable for the simultaneous screening for 41 viral and five bacterial agents related to respiratory tract infections, including the human coronaviruses NL63, HKU1, OC43, 229E, Middle East respiratory syndrome coronavirus, SARS-CoV, and SARS-CoV-2. RespiCoV was applied to 150 patient samples with suspected SARS-CoV-2 infection and compared with specific real-time PCR. Additionally, several respiratory tract pathogens were identified in samples tested positive or negative for SARS-CoV-2. Finally, RespiCoV was experimentally compared to the commercial RespiFinder 2SMART multiplex screening assay (PathoFinder, The Netherlands).

## Introduction

Infections of the respiratory tract range from the mild, self-limiting common cold to life-threatening illnesses and epidemics caused by influenza viruses, severe acute respiratory syndrome coronavirus (SARS-CoV), or Middle East respiratory syndrome coronavirus (MERS) [1, 2]. Recently, the severe acute respiratory syndrome coronavirus 2 (SARS-CoV-2) has been causing an ongoing global pandemic with more than 281,808,270 diagnosed cases of Covid-19

**Data Availability Statement:** All sequence data was deposited under the project number PRJEB49379 at the European Nucleotide Archive.

**Funding:** The author(s) received no specific funding for this work.

**Competing interests:** The authors have declared that no competing interests exist.

and 5,411,759 deaths as of 29 December 2021 (https://covid19.who.int/). Shortly after the identification and whole-genome sequencing of the novel emerging virus, specific real-time PCRs for SARS-CoV-2 diagnostics have been developed and deployed extensively [3–6]. Although up-to-date, fast, reliable, and specific real-time PCR-based SARS-CoV-2 diagnostics has the highest priority for control and containment of the Covid-19 pandemic, the identification and possible relevance of viral or bacterial co-infections for the severity of the course of Covid-19 have been addressed in many studies [7–10]. During the SARS-CoV pandemic in 2003, reports on dual infections were scarce [11, 12]. However, it has been shown by a systematic review that 19% of patients with COVID-19 have bacterial and viral co-infections which are associated with poorer outcomes [13]. As patients with symptoms described for Covid-19 are usually exclusively tested for SARS-CoV-2, many patients with negative results remain undiagnosed for co-infections, which can lead to non-specific treatment or incorrect treatment of hospitalized patients as well as uncertainty regarding the patients' health status and a needless placing in quarantine.

In contrast to specific real-time PCRs, Illumina and Nanopore shotgun sequencing enable the unbiased detection of one or several pathogens simultaneously from sputum or swab samples and have previously been performed for identification of respiratory tract pathogens, including *Streptococcus pneumoniae* and influenza virus [14–16]. However, shotgun sequencing generates a high amount of data accompanied by high costs, involuntary sequencing of the hosts' DNA which conflicts with personal data protection, and low sensitivity of virus identification.

Here we present an amplicon-based MinION sequencing approach (referred to as RespiCoV) with 114 primers for simultaneous diagnostics of SARS-CoV-2 and further 40 viral and five bacterial agents related to respiratory tract infections (Table 1). This approach can contribute to the detection of co-infections in patients infected with SARS-CoV-2 and aid in differential diagnostics of patients tested negative for SARS-CoV-2.

## Methods

### Primer design and evaluation

The targeted common upper respiratory tract viruses and bacteria for RespiCoV were chosen based on the publications by Hodinka et al. [17] and Jain et al. [18]. Additionally, herpes simplex virus type 1 and Epstein-Barr virus have been associated with upper respiratory tract infections in critically ill patients, and viruses were included as targets of the RespiCoV assay [19, 20]. Varicella zoster virus and herpes simplex virus type 2 were included in the RespiCoV Panel for validation of the method and for amplification as sequencing controls. Because generic markers like 16S rRNA can identify bacteria, we included only the most prominent bacteria.

The 114 primers (S1 Table) for the RespiCoV Panel were designed by using Primer3 v2.3.7 (https://primer3.org/). For the primer design, all references for each virus and bacteria species in Table 1 were aligned by using MAFFT v7.450, and primers were placed on conserved regions. For SARS-CoV-2, primers were designed to target three genome regions of 374, 263, and 363 bp in length, respectively (bp regions 15,814–16,188 (ORF1ab); 16,821–17,084 (ORF1ab); 29,337–29,700 (N/ORF10); based on reference MT345888). These primers can detect also SARS-CoV, with significant differences in the amplified sequence enabling a clear discrimination between SARS-CoV and SARS-CoV-2 (90.6%, 91.6%, and 88.4% referred to BetaCoV/Germany/BavPat1/2020 [GISAID] und NC_004718.3). All primers have been selected based on their melting temperature Tm (minimum Tm 58˚C, maximum Tm 62˚C, optimal Tm 60˚C), resulting amplicon length (250–600 bp), and a minimum of possible hetero-dimer formation.

**Table 1. List of viruses and bacteria targeted by the RespiCoV Panel.**

| Family | Genus | Species |
|---|---|---|
| *Adenoviridae* | *Mastadenovirus* | Human adenovirus A |
| | | Human adenovirus B |
| | | Human adenovirus C |
| | | Human adenovirus D |
| | | Human adenovirus E |
| | | Human adenovirus F |
| *Coronaviridae* | *Betacoronavirus* | SARS coronavirus |
| | | SARS coronavirus 2 |
| | | MERS coronavirus |
| | | Human coronavirus OC43 |
| | | Human coronavirus HKU1 |
| | *Alphacoronavirus* | Human coronavirus NL63 |
| | | Human coronavirus 229E |
| *Hantaviridae* | *Orthohantavirus* | Hantaan virus |
| *Herpesviridae* | *Cytomegalovirus* | Human cytomegalovirus |
| | *Lymphocryptovirus* | Epstein-Barr virus |
| | *Simplexvirus* | Herpes simplex virus 1 |
| | | Herpes simplex virus 2 |
| | *Varicellovirus* | Varicella zoster virus |
| *Orthomyxoviridae* | *Influenzavirus A* | Influenza A virus |
| | *Influenzavirus B* | Influenza B virus |
| | *Influenzavirus C* | Influenza C virus |
| *Paramyxoviridae* | *Respirovirus* | Human respirovirus 1 |
| | | Human respirovirus 3 |
| | *Rubulavirus* | Human respirovirus 2 |
| | | Human respirovirus 4 |
| | | Mumps orthorubulavirus |
| | *Henipavirus* | Nipah virus |
| | | Hendra virus |
| | *Morbillivirus* | Measles virus |
| *Parvoviridae* | *Bocaparvovirus* | Human bocavirus |
| *Picornaviridae* | *Enterovirus* | Rhinovirus A |
| | | Rhinovirus B |
| | | Rhinovirus C |
| | | Enterovirus A |
| | | Enterovirus B |
| | | Enterovirus C |
| | | Enterovirus D |
| | *Parechovirus* | Human parechovirus |
| *Pneumoviridae* | *Orthopneumovirus* | Human respiratory syncytial virus |
| | *Metapneumovirus* | Human metapneumovirus |
| *Alcaligenaceae* | *Bordetella* | *Bordetella pertussis/parapertussis* |
| *Chlamydiaceae* | *Chlamydophila* | *Chlamydophila pneumoniae* |
| *Legionellaceae* | *Legionella* | *Legionella pneumophila* |
| *Mycoplasmataceae* | *Mycoplasma* | *Mycoplasma pneumoniae* |
| *Streptococcaceae* | *Streptococcus* | *Streptococcus pneumoniae/pseudopneumoniae* |

## Panel evaluation with human clinical specimens

The performance of the RespiCoV Panel was first tested on clinical specimens from 12 patients with a clinical diagnosis of a respiratory tract infection (throat and nose swabs, sampled before the start of the SARS-CoV-2 pandemic). For comparison, samples were tested with the Patho-Finder RespiFinder 2SMART (PathoFinder B.V., Maastricht, The Netherlands), which is a multiplex real-time PCR system for identification of 18 viral and five bacterial pathogens (influenzavirus A and B, human parainfluenza virus types 1–4, human respiratory syncytial virus types A and B, human metapneumovirus, rhinovirus/enterovirus, bocavirus, adenovirus, coronavirus NL63, HKU1, 229E, OC43, *Mycoplasma pneumoniae*, *Legionella pneumophila*, *Chlamydophila pneumoniae*, and *Bordetella pertussis*). Furthermore, throat swabs from 150 patients with suspected SARS-CoV-2 infection were screened with the RespiCoV Panel and specific SARS-CoV-2 real-time PCR [6]. Patient samples were extracted with the Qiagen Viral RNA Mini Kit (Qiagen, Hilden, Germany). For the RespiCoV Panel, cDNA synthesis was performed according to the SuperScript IV Reverse Transcriptase protocol (Thermo Fisher Scientific, Waltham, MA, USA) with random hexamers (65˚C for 5 min and 23˚C for 10 min), followed by incubation at 55˚C for 10 min and inactivation at 80˚C for 10 min.

## Panel evaluation via samples for an international quality assurance exercise

The RespiCoV method was further tested with samples provided by INSTAND e.V. for a national quality assurance exercise. INSTAND e.V. offers exercises for quality assurance for medical laboratories across Germany. This exercise focused on the genomic detection of SARS-CoV-2 and contained inactivated samples for sensitivity (SARS-CoV-2 in different concentrations) and specificity (other coronaviruses).

For comparison, the samples were identified with specific real-time PCRs for SARS-CoV, SARS-CoV-2, MERS-CoV, OC43, NL63, 229E, and HKU1.

## PCR amplification

The patient samples were amplified in a single reaction with the following PCR conditions: 3 µl of viral cDNA, 1.6 µl of primer pool, 0.2 mM dNTP (Invitrogen, Karlsruhe, Germany), 4 µl of 10 x Platinum Taq buffer, 2 mM $MgCl_2$, and 5 U Platinum Taq polymerase (Invitrogen) with added water to a final volume of 25 µl. Cycling conditions were 94˚C for 5 min, 45 amplification cycles at 94˚C for 20 s, 65˚C for 30 s, 72˚C for 20 s, and a final extension step for 5 min (at 72˚C). Thermal cycling was performed in an Eppendorf Mastercycler Pro (Eppendorf Vertrieb Deutschland, Wesseling-Berzdorf, Germany) with a total runtime of 64 min.

## Library preparation and NGS sequencing

Amplified samples were processed for nanopore sequencing on the MinION (Oxford Nanopore Technologies, Oxford, United Kingdom). The libraries were prepared by using the ligation sequencing kit 1D, SQK-LSK109 (Oxford Nanopore Technologies). For combined sequencing of several samples on one flow cell, samples were barcoded with the Native Barcoding Expansion Kit (EXP-NBD104 and EXP-NBD114). Subsequently, the libraries were loaded onto Oxford Nanopore MinION SpotON Flow Cells Mk I, R9.4.1. (Oxford Nanopore Technologies). Samples were run for at least 30 min.

## Bioinformatics analysis

The Fast5 data generated during sequencing was transcribed to FastQ sequences by using Guppy v.3.4.5 (Oxford Nanopore Technologies) on the MinION IT device (MNT-001).

Computational separation of the barcoded samples was performed with Guppy v.3.4.5 for Windows. FastQ files for each sample were aligned to the reference sequences with Guppy v.4.0.11 for Linux and the resulting alignments were used for read counts. Primer sequences were soft clipped with bamclipper v.1.1.1 and all soft clippings from the BAM file were removed with custom python scripts. For species identification, consensus sequences generated from the reference alignments (Geneious prime v2020.2.3) were validated using online blast. As read counts can differ between runs, samples were only rated positive when the following parameters were met: number of total reads for each sample > 0.5% of the total reads from the run, number of reads for SARS-CoV-2 > 0.5% of all total reads of SARS-CoV-2 from the run plus the reads of SARS-CoV-2 identified in the negative control, and number of reads for SARS-CoV-2 > 50.

### Ethics statement

The studies involving human participants were reviewed and approved by the Ärztekammer Berlin (Berlin Medical Association; #Eth 20/40). The patients/participants provided their written informed consent to participate in this study.

## Results

### Comparison of RespiFinder 2SMART and the RespiCoV Panel

In one of the samples tested negative with the RespiFinder 2SMART, herpes simplex virus type 1 could be identified with the RespiCoV Panel (67,321 specific amplicons), which is not targeted by the RespiFinder 2SMART. Furthermore, in three of the patient samples tested positive with the RespiFinder 2SMART, additional pathogens could be identified with the RespiCoV Panel. *Streptococcus pneumoniae*, which is not targeted by the RespiFinder 2SMART, could be identified additionally in two of the samples, and Rhinovirus A could be identified in one of the samples. For three of the samples identified as positive with both methods, additional species/strain information could be gained by the sequence information obtained with the RespiCoV Panel. For example, human adenovirus could be specified further to human adenovirus type B and the lineage of influenzavirus B could be identified as Yamagata. For two of the samples tested positive with both methods, read numbers after MinION sequencing were very low (55 reads for Human respiratory syncytial virus B and 676 reads for Human metapneumovirus). For the remaining samples, 10,937–192,431 reads were sequenced in one hour, providing sufficient viral reads for identification within the first minutes of sequencing (Table 2).

### Screening of samples from patients with suspected SARS-CoV-2 infection with the RespiCoV Panel

Of the 150 clinical samples, 66 samples were identified as negative and 84 samples were identified as positive for SARS-CoV-2 with a specific SARS-CoV-2 real-time PCR in our routine diagnostics (Cq range of 18–38, Table 3). With RespiCoV, 65 of the 66 negative samples were correctly identified as negative for SARS-COV-2, whereas one sample was identified as positive for SARS-CoV-2 with low read numbers of SARS-CoV-2 amplicons after sequencing (n = 4000, mean read numbers for samples within Cq range 18–28: 35,000; and 19,000 within Cq range 29–33). However, the patient had been tested negative by specific real-time PCR previously, but after a series of positive tests.

Of the 84 samples tested positive by specific real-time PCR, 35 samples were within the Cq range of 18–28, 24 within a Cq range of 29–33, and 25 within a Cq range of 34–38. Of the 35 samples tested positive for SARS-CoV-2 by specific qPCR within a Cq range of 18–28, all

**Table 2. Comparison of results for screening of samples from patients with respiratory tract infections with the RespiFinder 2SMART and the RespiCoV Panel.**
Results for the RespiFinder 2SMART could be confirmed with the RespiCoV Panel. Some additional species could be identified (*Streptococcus pneumoniae*, herpes simplex virus 1, and Rhinovirus A).

| ID | RespiCoV | RespiFinder | Reads after ~1 h | Target reads / total reads |
|---|---|---|---|---|
| 1 | Influenza B virus (Yamagata) | Influenza B virus | 38,132 | 0.23 |
| | *Streptococcus pneumoniae* | negative (not included) | 126,832 | 0.75 |
| 2 | Herpes simplex virus 1 | negative (not included) | 67,321 | 0.96 |
| 3 | Influenza A virus H3N2 | Influenza A virus H3N2 | 90,898 | 0.94 |
| 4 | Coronavirus OC43 | Coronavirus OC43 | 118,726 | 0.98 |
| 5 | negative | Negative | - | |
| 6 | Influenza A virus H1N1 | Influenza A virus H1N1 | 132,348 | 0.86 |
| 7 | Human metapneumovirus | Human metapneumovirus | 676 | 0.01 |
| | *Streptococcus pneumoniae* | negative (not included) | 110,647 | 0.97 |
| 8 | Human bocavirus | Human bocavirus | 22,008 | 0.91 |
| 9 | Coronavirus 229E | Coronavirus 229E | 209,539 | 0.98 |
| 10 | negative | negative | | |
| 11 | Human adenovirus B | Human adenovirus | 74,003 | 0.26 |
| | Human parainfluenza virus type 2 | Human parainfluenza virus type 2 | 192,431 | 0.67 |
| | Rhinovirus A | negative | 10,937 | 0.04 |
| 12 | Human respiratory syncytial virus B | Human respiratory syncytial virus | 55 | 0.06 |

samples were identified correctly as positive for SARS-CoV-2 with RespiCoV. Although read numbers can differ between different runs (level of multiplexing, sequencing time, distribution of positive and negative samples, quality of flow cell), read numbers of SARS-CoV-2 after sequencing ranged from 2731 to 148,520 with a mean of 35,000 reads per sample. Of the 24 samples tested positive with a Cq range of 29–33 by specific real-time PCR, 19 samples could also be identified as positive with RespiCoV. Furthermore, for 4 of the samples of the Cq range 29–32, low reads of SARS-CoV-2 were identified (349–418 reads), but read numbers were below the threshold for positive identification by RespiCoV. One of the 24 samples identified as positive by specific real-time PCR (Ct 29–33) was identified as clearly negative (Cq 31), with only 19 reads of SARS-CoV-2 after RespiCoV PCR and sequencing. For samples identified as positive by specific real-time PCR with Cq values between 34 and 38, only 3 of 25 samples were identified as positive with RespiCoV.

**Table 3. Results of the RespiCoV analyses for 150 samples from patients with suspected SARS-CoV-2 infection (negative = 66, positive = 84).** Sequences of 32 pathogens other than SARS-CoV-2 could be identified, with 23 pathogens in 17 of the 66 negative samples. Co-infections were identified in 8 of the SARS-CoV-2-positive samples.

| | Negative | Cq 18–28 | Cq 29–33 | Cq 34–38 |
|---|---|---|---|---|
| | n = 66 | n = 35 | n = 24 | n = 25 |
| SARS-CoV-2 pos. | 1 | 35 | 19 | 3 |
| SARS-CoV-2 neg. | 65 | 0 | 5 | 22 |
| *Streptococcus pneumoniae* | 2 | 2 | 1 | 0 |
| Epstein-Barr virus | 6 | 1 | 1 | 1 |
| Human cytomegalovirus | 4 | 0 | 0 | 0 |
| Herpes simplex virus 1 | 2 | 1 | 1 | 0 |
| Human adenovirus B1 | 3 | 0 | 0 | 0 |
| Rhinovirus A | 3 | 0 | 0 | 0 |
| Rhinovirus B | 2 | 0 | 0 | 0 |

As shown in Fig 1, there is a good correlation between virus genome load represented by the Cq value and the read number within one sequencing run, but not between different runs (shown for three different runs).

In the 150 samples, sequences of 32 pathogens other than SARS-CoV-2 could be identified, with 23 pathogens in 17 of the 66 negative samples (25%) (*Streptococcus pneumoniae*, Epstein-Barr virus, human cytomegalovirus, herpes simplex virus 1, human adenovirus B, and rhinovirus A and B). Double and triple infections were identified in 5 samples (*Streptococcus pneumoniae* and Epstein-Barr virus; Epstein-Barr virus, human cytomegalovirus, and herpes simplex virus; Epstein-Barr virus and human cytomegalovirus in two samples; *Streptococcus pneumoniae* and Rhinovirus B). Within the 84 SARS-CoV-2-positive samples, only seven co-infections were identified in six samples (*Streptococcus pneumoniae*, Epstein-Barr virus, herpes simplex virus 1) with one co-infection of *Streptococcus pneumoniae* and Epstein-Barr virus in one sample.

## Evaluation on INSTAND external quality assurance exercise samples

The RespiCoV method was further tested with samples provided by INSTAND e.V. for quality assurance of SARS-CoV-2 diagnostics in medical laboratories across Germany. The results obtained with RespiCoV were identical when compared with the real-time PCR results (Table 4). After 30 min of sequencing, 137,295 reads were obtained from the samples with high SARS-CoV-2 concentration (Cq 21.4). For the samples with a low concentration of SARS-CoV-2, only 4,657 reads were sequenced, but the read number was sufficient for identification of the virus within the first minutes of sequencing.

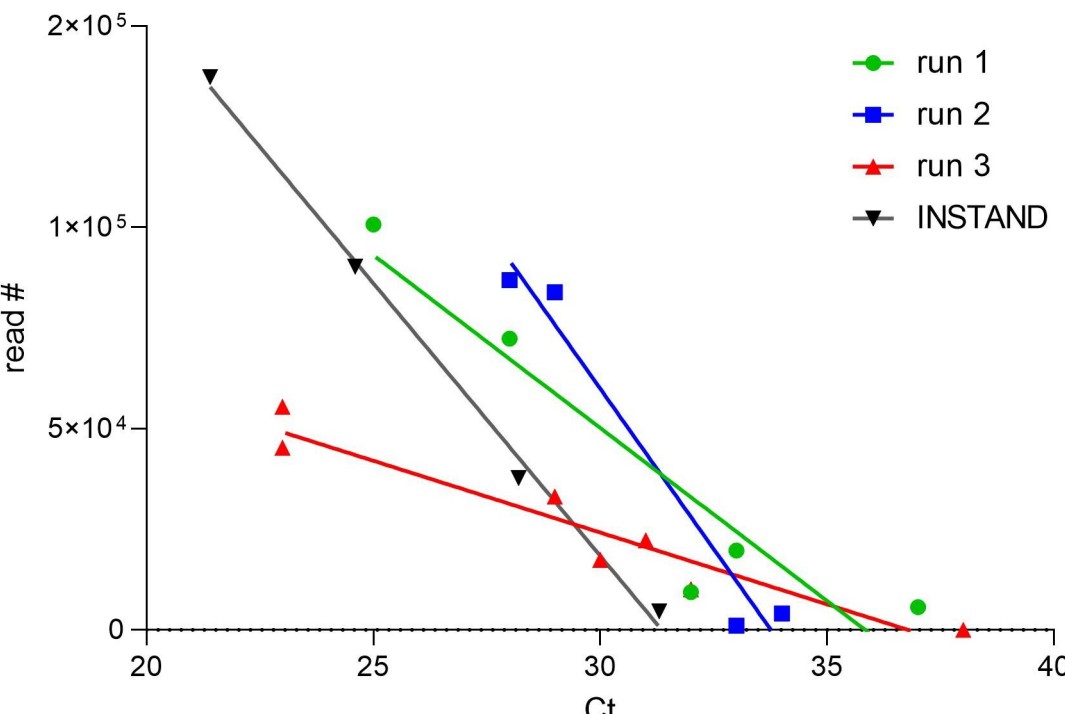

**Fig 1. Correlation of SARS-CoV-2-specific reads obtained by RespiCoV (shown for three different sequencing runs) and the Cq value generated by a specific real-time PCR assay.** Even if accurate quantification by RespiCoV is questionable, within one run the correlation is significant (R squared: 0.88, 0.96, 0.91, and 0.99 for run 1, run 2, run 3, and the INSTAND samples, respectively).

**Table 4. Results of RespiCoV and specific real-time PCRs of the INSTAND exercise for quality assurance for SARS-CoV-2 diagnostics.**

| Sample | Result real-time PCR | Cq real-time PCR | RespiCoV | Amplicon reads ~ 30 min | Target reads / total reads |
|---|---|---|---|---|---|
| INSTAND-340059 | SARS-CoV-2 | 21.4 | SARS-CoV-2 | 137,295 | 0.94 |
| INSTAND-340060 | OC43 | 25.3 | OC43 | 23,063 | 0.93 |
| INSTAND-340061 | SARS-CoV-2 | 31.3 | SARS-CoV-2 | 4,657 | 0.93 |
| INSTAND-340062 | negative | | negative | | |
| INSTAND-340063 | SARS-CoV-2 | 24.6 | SARS-CoV-2 | 90,265 | 0.93 |
| INSTAND-340064 | SARS-CoV-2 | 28.2 | SARS-CoV-2 | 37,722 | 0.94 |
| INSTAND-340065 | 229E | 24.7 | 229E | 46,051 | 0.95 |
| NK | negative | | negative | | |

## Discussion

In this study, we introduce an amplicon-based MinION sequencing approach, referred to as RespiCoV, which is able to identify and differentiate 41 viral and five bacterial species related to respiratory tract infections, including the human coronaviruses 229E, HKU1, NL63, OC43, MERS, SARS-CoV, and SARS-CoV-2, the latter challenging the world in an ongoing pandemic since 2019. We could show that the RespiCoV Panel is able to identify several viral and bacterial species in patients with symptoms of respiratory tract infections. Furthermore, samples from patients with diagnosed infections with SARS-CoV-2 were identified with the RespiCoV Panel, even if viral load was low (up to a Cq value of 33). Although the identification was not shown experimentally for all viral and bacterial targets of the RespiCoV Panel, the performance of the method was shown for several pathogens, including influenza A virus, influenza B virus, human coronavirus OC43 and 229E, human adenovirus B, human bocavirus, human metapneumovirus, human respiratory syncytial virus, human parainfluenza virus types 2, herpes simplex virus type 1, *S. pneumoniae*, and SARS-CoV-2. Compared with the extensively used and validated RespiPanel 2SMART, we could show that the RespiCoV Panel can be used as an approach for the simultaneous identification of respiratory tract pathogens. Just in one case, only low read numbers of Human respiratory syncytial virus could be identified with the RespiCoV Panel, which may be the result of low virus concentration or the primer design, that could be adapted by integrating additional primers into the RespiCoV primer pool.

Although reliable, fast, and accurate real-time PCR is the gold standard for SARS-CoV-2 detection, the method described here can further contribute to the diagnostics and differential diagnostics of patients with symptoms described for Covid-19. Identification of viral and bacterial co-infections has been performed in several studies with real-time PCR, but the abundance and potential impact of these infections remained unknown. In the 2009 H1N1 influenza outbreak, co-infections of patients with H1N1 and a second respiratory virus were associated with an increased risk of complications [21]. Furthermore, in children co-infections with respiratory syncytial virus and metapneumovirus or rhinovirus were associated with a 10-fold greater risk of Pediatric Intensive Care Unit level of care [22, 23]. In contrast, other studies have found less severe clinical outcomes with viral co-infection or showed no correlation of co-infections and severity of disease [24, 25]. Co-infections of patients diagnosed for SARS-CoV-2 identified by specific real-time PCRs performed in two independent studies also included common respiratory viruses (influenza A virus, rhinovirus, human respiratory syncytial virus, human coronavirus HKU1, human parainfluenzavirus type 1, and human metapneumovirus), but infection rates were low (5.8% and 3.2%, respectively) [7, 9, 26]. Another study reported 22.4% of all patients assigned to the emergency department to be infected with both SARS-CoV-2 and a second viral pathogen (Editor's note in [27]).

In our study, for some of the samples diagnosed as positive for SARS-CoV-2 by specific real-time PCR and the RespiCoV Panel, viral co-infections with herpes simplex virus type 1 and Epstein-Barr virus could be identified, both of which are usually not included in screening of patients with respiratory tract infections. However, herpes simplex virus type 1 infection or reactivation in the lower and upper respiratory tract has been recorded in patients in intensive care and has increasingly been associated with pulmonary diseases with poor outcome [19, 28]. Although quantification with the RespiCoV Panel is not validated, low read numbers of herpes simplex virus type 1 and Epstein-Barr virus indicate low viral concentration in the throat.

Furthermore, 24.7% of patients infected with H1N1 during the influenza pandemic showed co-infection with bacteria, mainly *Staphylococcus aureus* and *Streptococcus pneumoniae* [29]. *S. pneumoniae* has also been identified as a co-infection in patients infected with influenza during the pandemic 1918–1919 and during the Asian and Hong Kong influenza pandemics of 1957 and 1968 [30, 31].

In direct comparison, the RespiCoV Panel was shown to be less sensitive than specific real-time PCRs for SARS-CoV-2, but able to identify SARS-CoV-2 from patient samples with a Cq up to 33. Hands-on and sequencing take several hours and costs can be higher than commercial multiplex-PCR; however, additional information about the identified pathogen, including species and strain, can be obtained by the method. Due to the generation of specific amplicons, no sequence information of the host is generated which could be conflicting with personal data protection for shotgun sequencing.

## Conclusion

Since the ongoing outbreak of SARS-CoV-2 starting in 2019, specific real-time PCR diagnostics has been contributing to the elucidation and containment of the pandemic. However, differential diagnostics and identification of Covid-19 co-infections might contribute to health care management and provide further understanding of Covid-19 courses of diseases. With RespiCoV, we have introduced an approach of highly multiplexed PCRs and MinION sequencing which can be used for rapid and comprehensive simultaneous screening for many pathogens.

## Supporting information

**S1 Table. Primer sequences for the RespiCoV Panel.**
(XLSX)

## Acknowledgments

We kindly thank Ursula Erikli for copy-editing and Ute Kramer for sequencing.

## Author Contributions

**Conceptualization:** Annika Brinkmann, Lars Schaade, Andreas Nitsche.

**Data curation:** Annika Brinkmann.

**Formal analysis:** Steven Uddin, Sophie-Luisa Ulm, Katharina Pape, Sophie Förster, Khalid Enan, Jalal Nourlil, Eva Krause, Janine Michel.

**Writing – original draft:** Annika Brinkmann, Andreas Nitsche.

**Writing – review & editing:** Andreas Nitsche.

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
