## [Decision Letter · Decision Letter 0]

7 Dec 2021

PONE-D-21-32371RespiCoV: Simultaneous identification of Severe Acute Respiratory Syndrome Coronavirus 2 (SARS-CoV-2) and 40 respiratory tract viruses by amplicon-based Oxford-Nanopore MinION sequencingPLOS ONE

Dear Dr. Brinkmann,

Thank you for submitting your manuscript to PLOS ONE. After careful consideration, we feel that it has merit but does not fully meet PLOS ONE’s publication criteria as it currently stands. Therefore, we invite you to submit a revised version of the manuscript that addresses the points raised during the review process.

We look forward to receiving your revised manuscript.

Kind regards,

Kok Keng Tee, Ph.D.

Academic Editor

PLOS ONE

Journal Requirements:

Reviewers' comments:

Reviewer's Responses to Questions

**Comments to the Author**

1. Is the manuscript technically sound, and do the data support the conclusions?

Reviewer #1: Yes

Reviewer #2: Yes

2. Has the statistical analysis been performed appropriately and rigorously? 

Reviewer #1: N/A

Reviewer #2: N/A

3. Have the authors made all data underlying the findings in their manuscript fully available?

Reviewer #1: Yes

Reviewer #2: No

4. Is the manuscript presented in an intelligible fashion and written in standard English?

Reviewer #1: Yes

Reviewer #2: Yes

5. Review Comments to the Author

Reviewer #1: The authors describe a method based on the combination of multiplex PCR with MinION sequencing for the simultaneous detection of SARS-CoV-2 and a panel of potential co-infecting agents. The study was well conducted and no major issues were identified. Some minor comments are provided below.

Please provide additional information on how the species identification was performed upon acquisition of the reads.

Use the term "Cq" instead of "Ct" according to Bustin et al., throughout the manuscript, including in the tables.

In line 161 a "." seems to be missing in the Cq value.

Whenever referring to Tables and Figures, start with with capital "T" and/ or "F".

Tables should be improved, provide captions with additional information.

With the data obtained can the authors provide details regarding the analytical sensitivity, specificity and accuracy of their method?

Reviewer #2: Brinkmann and colleagues presented and evaluated an alternative method for simultaneous identification of respiratory pathogens using amplicon sequencing on a nanopore platform. The method was evaluated with clinical samples and compared with commercially available multiplex screening assays. In addition, the methodology was tested in an international quality assurance exercise.

The study is interesting and valuable because respiratory pathogens other than SARS-CoV-2 are often undiagnosed during the pandemic. Sequencing using the Nanopore platform is also becoming more available worldwide. The manuscript is concise and clearly written, although some minor improvements should be made.

Title: the proposed panel also is intended to detect some bacterial pathogens. It would be good to adjust the title accordingly to indicate the broader scope of this method.

Data access statement: the authors state that all data are available through the SRA databases. The accession number should be included in the manuscript.

Bioinformatics analysis: the author used the standard software provided by the sequencing company, which could be advantageous in a clinical setting, but the authors may wish to compare their results with more advanced workflows available for nanopore reads.

Please provide further details on the primer trimming method.

How do the authors prevent contamination to avoid false positive results?

Results:

Authors should include normalization of read counts (e.g., per million mapped reads) in the results to improve comparability across different runs.

Discussion:

The authors should include considerations of the time and cost of the workflow to provide a sense of the feasibility of the method in clinics.

6. PLOS authors have the option to publish the peer review history of their article (what does this mean?). If published, this will include your full peer review and any attached files.

Reviewer #1: No

Reviewer #2: No

---

## [Author Response · Author response to Decision Letter 0]

4 Jan 2022

Dear Reviewers, dear Editor,

Thank you for considering and reviewing our manuscript. Please find below our response.

AB: We have changed the style according to the PLOS ONE style templates and hope to meet all criteria. 

2. Please review your reference list to ensure that it is complete and correct.

AB: There have been no final changes to the reference list.

3. Please provide additional details regarding participant consent. 

We have added the Ethics statement to the manuscript (line 145-148)

4. Please include your full ethics statement in the ‘Methods’ section of your manuscript file. 

We have added the Ethics statement to the manuscript (line 145-148)

 

Comments to the Author

1. Is the manuscript technically sound, and do the data support the conclusions?

Reviewer #1: Yes

Reviewer #2: Yes

2. Has the statistical analysis been performed appropriately and rigorously? 

Reviewer #1: N/A

Reviewer #2: N/A

3. Have the authors made all data underlying the findings in their manuscript fully available?

Reviewer #1: Yes

Reviewer #2: No

AB: We have uploaded all data (PRJEB49379 at the European Nucleotide Archive)

4. Is the manuscript presented in an intelligible fashion and written in standard English?

Reviewer #1: Yes

Reviewer #2: Yes

5. Review Comments to the Author

Reviewer #1: The authors describe a method based on the combination of multiplex PCR with MinION sequencing for the simultaneous detection of SARS-CoV-2 and a panel of potential co-infecting agents. The study was well conducted and no major issues were identified. Some minor comments are provided below.

Please provide additional information on how the species identification was performed upon acquisition of the reads.

AB: Dear Reviewer #1, thank you for reviewing our manuscript. Species identification was performed by generating reference alignments with subsequent blast for validation. We have included additional information (line 136-137).

Use the term "Cq" instead of "Ct" according to Bustin et al., throughout the manuscript, including in the tables.

AB: We have changed this accordingly.

In line 161 a "." seems to be missing in the Cq value.

AB: We have corrected this (should have been a “-“)

Whenever referring to Tables and Figures, start with with capital "T" and/ or "F".

Tables should be improved, provide captions with additional information.

AB: We have corrected this and included additional information in the captions of Table 2 and 3.

With the data obtained can the authors provide details regarding the analytical sensitivity, specificity and accuracy of their method?

AB: We have included some general considerations in the discussion (line 284 – 290)

 

Reviewer #2: Brinkmann and colleagues presented and evaluated an alternative method for simultaneous identification of respiratory pathogens using amplicon sequencing on a nanopore platform. The method was evaluated with clinical samples and compared with commercially available multiplex screening assays. In addition, the methodology was tested in an international quality assurance exercise.

The study is interesting and valuable because respiratory pathogens other than SARS-CoV-2 are often undiagnosed during the pandemic. Sequencing using the Nanopore platform is also becoming more available worldwide. The manuscript is concise and clearly written, although some minor improvements should be made.

Title: the proposed panel also is intended to detect some bacterial pathogens. It would be good to adjust the title accordingly to indicate the broader scope of this method.

AB: Dear Reviewer #2, thank you for your suggestions. We have added “bacteria” To the title. 

Data access statement: the authors state that all data are available through the SRA databases. The accession number should be included in the manuscript.

AB: We have uploaded all data (PRJEB49379 at the European Nucleotide Archive) and included the data access statement in the methods section.

Bioinformatics analysis: the author used the standard software provided by the sequencing company, which could be advantageous in a clinical setting, but the authors may wish to compare their results with more advanced workflows available for nanopore reads.

AB: Although comparison of analysis workflows might be interesting, we think that this is not scope of the manuscript. As only short target amplicons need to be evaluated, aligning the reads to reference targets is the most simple and straightforward approach without the need for benchmarking.

Please provide further details on the primer trimming method.

AB: Added to the bioinformatics section.

How do the authors prevent contamination to avoid false positive results?

AB: All results are evaluated based on the negative control in the run (line 137-141) to monitor contamination. To avoid contaminations in the lab, we premise general precautions. However, theses are not specifically addressed in the manuscript.

Results:

Authors should include normalization of read counts (e.g., per million mapped reads) in the results to improve comparability across different runs.

AB: For comparability, we included the ratio of target reads / total reads in the sample (table 2, table 4).

Discussion:

The authors should include considerations of the time and cost of the workflow to provide a sense of the feasibility of the method in clinics.

AB: We have included some general considerations in the discussion (line 284 – 290)

---

## [Decision Letter · Decision Letter 1]

18 Feb 2022

RespiCoV: Simultaneous identification of Severe Acute Respiratory Syndrome Coronavirus 2 (SARS-CoV-2) and 46 respiratory tract viruses and bacteria by amplicon-based Oxford-Nanopore MinION sequencing

PONE-D-21-32371R1

Dear Dr. Brinkmann,

We’re pleased to inform you that your manuscript has been judged scientifically suitable for publication and will be formally accepted for publication once it meets all outstanding technical requirements.

Kind regards,

Kok Keng Tee, Ph.D.

Section Editor

PLOS ONE

Additional Editor Comments (optional):

Reviewers' comments:

Reviewer's Responses to Questions

**Comments to the Author**

1. If the authors have adequately addressed your comments raised in a previous round of review and you feel that this manuscript is now acceptable for publication, you may indicate that here to bypass the “Comments to the Author” section, enter your conflict of interest statement in the “Confidential to Editor” section, and submit your "Accept" recommendation.

Reviewer #1: All comments have been addressed

Reviewer #2: All comments have been addressed

2. Is the manuscript technically sound, and do the data support the conclusions?

Reviewer #1: Yes

Reviewer #2: Yes

3. Has the statistical analysis been performed appropriately and rigorously? 

Reviewer #1: (No Response)

Reviewer #2: N/A

4. Have the authors made all data underlying the findings in their manuscript fully available?

Reviewer #1: Yes

Reviewer #2: Yes

5. Is the manuscript presented in an intelligible fashion and written in standard English?

Reviewer #1: Yes

Reviewer #2: Yes

6. Review Comments to the Author

Reviewer #1: All the comments have been addressed and no further information is required. From this reviewer's point of view the manuscript is acceptable for publication.

Reviewer #2: Brinkmann and colleaques present a revised manuscript where all comments have been adressed carefully by the authors.

7. PLOS authors have the option to publish the peer review history of their article (what does this mean?). If published, this will include your full peer review and any attached files.

Reviewer #1: **Yes: **Alejandro Garrido-Maestu

Reviewer #2: No

---

## [Editor Report · Acceptance letter]

28 Feb 2022

PONE-D-21-32371R1 

RespiCoV: Simultaneous identification of Severe Acute Respiratory Syndrome Coronavirus 2 (SARS-CoV-2) and 46 respiratory tract viruses and bacteria by amplicon-based Oxford-Nanopore MinION sequencing 

Dear Dr. Brinkmann:

I'm pleased to inform you that your manuscript has been deemed suitable for publication in PLOS ONE. Congratulations! Your manuscript is now with our production department. 

Kind regards, 

on behalf of

Dr. Kok Keng Tee 

Section Editor

PLOS ONE